# Calmodulin as a Key Regulator of Exosomal Signal Peptides

**DOI:** 10.3390/cells12010158

**Published:** 2022-12-30

**Authors:** Kenji Ono, Mikio Niwa, Hiromi Suzuki, Nahoko Bailey Kobayashi, Tetsuhiko Yoshida, Makoto Sawada

**Affiliations:** 1Department of Brain Function, Division of Stress Adaptation and Protection, Research Institute of Environmental Medicine, Nagoya University, Nagoya 464-8601, Aichi, Japan; 2Department of Molecular Pharmacokinetics, Graduate School of Medicine, Nagoya University, Nagoya 464-8601, Aichi, Japan; 3Institute for Advanced Sciences, Toagosei Co., Ltd., Tsukuba 300-2611, Ibaraki, Japan

**Keywords:** signal peptide, exosomes, intercellular communication, calmodulin

## Abstract

Signal peptides (SPs) and their fragments play important roles as biomarkers and substances with physiological functions in extracellular fluid. We previously reported that SP fragments were released into extracellular fluid via exosomes and bound to calmodulin (CaM), an exosomal component, in a cell-free system. However, it currently remains unclear whether CaM intracellularly interacts with SP fragments or is involved in the trafficking of these fragments to exosomes. Therefore, the present study examined the binding of CaM to SP fragments in T-REx AspALP cells, transformed HEK293 cells expressing amyloid precursor protein (APP) SP flanking a reporter protein, and their exosomes. APP SP fragments were detected in exosomes from T-REx AspALP cells in the absence of W13, a CaM inhibitor, but were present in lower amounts in exosomes from W13-treated cells. Cargo proteins, such as Alix, CD63, and CD81, were increased in W13-treated T-REx AspALP cells but were decreased in their exosomes. Furthermore, CaM interacted with heat shock protein 70 and CD81 in T-REx AspALP cells and this increased in the presence of W13. APP SP fragments were detected in intracellular CaM complexes in the absence of W13, but not in its presence. These results indicate that CaM functions as a key regulator of the transport of SP fragments into exosomes and plays novel roles in the sorting of contents during exosomal biogenesis.

## 1. Introduction

Signal peptides (SPs) are sequences at the N terminus of newly synthesized proteins that are important for targeting the endoplasmic reticulum (ER). SPs are removed from synthesized proteins on the ER membrane by signal peptidases [1] and are further cleaved by signal peptide peptidases into two fragments [2]. Although cleaved fragments were previously considered to be degraded intracellularly [3], some SPs and their fragments have been identified in extracellular fluid, indicating physiological functions. SP fragments from A-, B-, and C-type natriuretic peptides are present in human circulation and are altered by myocardial infarction. Three SP types derived from natriuretic peptides secreted from the heart have been identified, and the measurement of SPs in plasma is expected to be useful as biomarkers for cardiovascular diseases [4,5,6]. Furthermore, cleaved SPs from protease-activated receptor 1 have been shown to not only suppress ocular neovascularization and inflammation [7], but also contribute to cardio- and renal protection after ischemia and reperfusion injury [8,9]. The surface trafficking and synaptic targeting of GluK1, a kainite-type glutamate receptor that plays an important role in the excitatory transmission and synaptic plasticity, are both repressed by GluK1 SP through direct interaction with the amino-terminal domain, indicating that cleaved SPs behave as a ligand of GluK1 to repress the forward trafficking of the receptor [10]. Although these findings indicate that SPs and cleaved SPs are released into extracellular fluid via various routes without being degraded, the underlying mechanisms have not yet been elucidated in detail.

We recently reported that SPs were secreted from cells via exosomes. A C-terminal SP fragment of amyloid precursor protein (APP), a type I membrane protein, was secreted into extracellular fluid via exosomes using transformed HEK293 (T-REx AspALP) cells expressing APP SP flanking a reporter protein. An N-terminal SP fragment of human placental secreted alkaline phosphatase (SEAP), a type II membrane protein [11], was detected in exosomes secreted from HEK-Blue hTLR3 cells, which expressed human Toll-like receptor 3 and an inducible SEAP. When HEK-Blue hTLR3 cells were stimulated with a TLR3 ligand, the synthesis of SEAP was induced, and SEAP SP fragments in exosomes increased in parallel with SEAP secretion in a TLR3 ligand concentration-dependent manner [12]. Exosomes are extracellular vesicles that are essential for intercellular communication. They are formed within intracellular multivesicular endosomes and encapsulate biofunctional molecules, such as proteins, peptides, and miRNAs. SP fragments appear to be transported from the ER to endosomes without being degraded. We demonstrated that APP SP fragments bound to calmodulin (CaM) In a cell-free system [12] and a previous study showed that preprolactin SP fragments interacted with CaM [13]. CaM is a ubiquitous Ca^2+^-binding protein that serves as a physiological effector of a wide range of biological processes, such as immune responses, metabolism, higher brain functions, and intracellular migration [14]. CaM is also an exosomal component, as are cargo proteins, such as Alix, heat shock protein 70 (HSP70), and tetraspanins [15]. Interactions between CaM and HSP70 have been shown to regulate the cell cycle and apoptosis in human cells [16]. Although the mechanisms by which these proteins encapsulate into exosomes currently remain unclear, the interactions between CaM and cargo proteins, such as HSP70, appear to be important for the sorting of contents during exosomal biogenesis.

In the present study, APP SP in exosomes from T-REx AspALP cells decreased in the presence of W13, a CaM inhibitor. The treatment with W13 also induced increases in the expression of some cargo proteins in cells and decreased their contents in exosomes. CaM formed a complex with HSP70 and CD81, and this complex increased in W13-treated T-REx AspALP cells. In contrast, APP SP decreased in this complex in W13-treated T-REx AspALP cells, suggesting that CaM mediated the transportation of SP fragments from the ER to exosomes by interacting with HSP70 and CD81.

## 2. Materials and Methods

### 2.1. Cells

T-REx Mock and T-REx AspALP cells were used in the present study. T-REx Mock and T-REx AspALP cells are HEK-293 cells that stably express the tetracycline repressor in the T-REx System (Thermo Fisher Scientific, Waltham, MA, USA) [17]. In brief, the APP SP-SEAP sequence, in which the SP sequence of SEAP was replaced with that of APP, was inserted into multicloning sites of the pcDNA 4/TO vector, and T-REx AspALP cells, into which the modified vector was genetically transferred, were selected as a stable mutant strain. T-REx Mock cells, into which the unmodified pcDNA 4/TO vector was genetically transferred, were generated as a stable strain. One million cells were plated on a 100 mm Falcon cell culture dish (Corning Inc., Corning, NY, USA) and cultured in Dulbecco’s Modified Eagle’s Medium (DMEM) (Sigma-Aldrich, St Louis, MO, USA) supplemented with 10% fetal bovine serum. Extracellular vesicles were removed by centrifugation at 11,000× *g* for 24 h followed by an incubation in 1 μg/mL doxycycline (Takara Bio Inc., Kusatsu Shiga, Japan) and Penicillin-Streptomycin (Thermo Fisher Scientific) with or without 20 μM W13 (N-(4-Aminobutyl)-5-chloronaphthalene-2-sulfonamide Hydrochloride), a CaM inhibitor (Tokyo Chemical Industry Co., Ltd., Tokyo, Japan), at 37 °C for 72 h in 5% CO_2_/95% humidified air. W13 concentrations were measured with reference to a previous study [18]. Photographs of cells were taken using a microscope equipped with a DP73 camera (Olympus, Tokyo, Japan). Conditioned medium and cells were collected in one dish and the number and viability of cells were assessed by trypan blue dye exclusion assay using the Countess II FL Automated Cell Counter (Thermo Fisher Scientific). To confirm whether apoptosis occurred, a part of the cells was stained with Alexa Fluor 488 conjugated Annexin V (Thermo Fisher Scientific) and cell apoptosis was analyzed using a CytoFLEX flow cytometer (Beckman Coulter, Brea, CA, USA). Cells were washed with phosphate-buffered saline (PBS) twice and stored as cell pellets at −80 °C.

### 2.2. Isolation of Exosomes from Conditioned Medium

Conditioned medium from one culture dish was centrifugated at 300× *g* at 4 °C for 5 min to remove live cells, and the supernatant was then centrifugated at 2000× *g* for 20 min to remove apoptotic vesicles. Microvesicles were removed by centrifugation at 10,000× *g* for 60 min. To enrich exosomes, the supernatant was concentrated using a Vivaspin 20 (100k) ultrafiltration spin column (Sartorius AG, Göttingen, Germany). Exosomes were isolated from the concentrated supernatant using a qEV 35 nm column (IZON Science Ltd., Christchurch, New Zealand) and Automatic Fraction Collector V2 instrument (IZON Science Ltd.) and collected in 800 μL of PBS.

### 2.3. Measurement of SEAP Activity

SEAP activity in a conditioned medium was measured using QUANTI-Blue Solution (InvivoGen, San Diego, CA, USA). In brief, 20 μL of conditioned medium was mixed with 180 μL of QUANTI-Blue Solution and incubated at 37 °C for 1 h in a 96-well plate. Absorbance at 620 nm was measured using a microplate reader.

### 2.4. Nanoparticle Tracking Analysis (NTA)

The number and average size of exosomes were measured using a NanoSight NS300 (Malvern Panalytical Ltd., Malvern, UK). Exosomes were diluted at 1:100 in degassed water to a final volume of 600 μL and applied through a syringe for measurement. The camera level was increased until all particles were distinctly visible without exceeding a particle signal saturation of over 20% (level 14–15). Automatic settings for the maximum jump distance and blur settings were utilized. The detection threshold was 5. In each measurement, five 60-s videos were captured under the following conditions: cell temperature, 25 °C; syringe pump speed, 100 (instrument-specific unit); camera, sCMOS; laser, 488 nm blue. After capturing, the number and size of exosomes were analyzed using NanoSight NTA 3.2 software build 3.2.16. Released exosomes (particles per cell) were calculated using data from the NTA and cell counting.

### 2.5. Western Blotting

Cells and exosomes from T-REx Mock and T-REx AspALP cells were lysed in RIPA buffer (20 mM Tris-HCl, 150 mM NaCl, 1 mM Na_2_EDTA, 1 mM EGTA, 1% NP-40, 1% sodium deoxycholate, 2.5 mM sodium pyrophosphate, 1 mM β-glycerophosphate, 1 mM Na_3_VO_4_, and 1 μg/mL leupeptin, pH 7.5) (Cell Signaling Technology, Danvers, MA, USA) by sonication in iced water. The BCA Protein Assay (Thermo Fisher Scientific, Rockford, IL, USA) was performed to measure protein concentrations. Cells were mixed with loading buffer with or without dithiothreitol (Cell Signaling Technology) to 2 mg/mL and exosomes to 1 mg/mL. The mixture was boiled for 5 min and 5 μL of the mixture was separated by SDS-PAGE on e-PAGELs (Atto, Tokyo, Japan). Separated proteins were then transferred to polyvinylidene difluoride membranes on an iBlot2 Gel Transfer Device (Thermo Fisher Scientific) and blocked in TBS (Tris-buffered saline) supplemented with 5% nonfat dry milk (Cell Signaling Technology) and 0.1% Tween 20 at room temperature for 1 h. Membranes were incubated in the presence of primary antibodies diluted in Can Get Signal Immunoreaction Enhancer Solution 1 (Toyobo, Osaka, Japan) at a 1:8000 dilution at 4 °C overnight. Primary antibodies were directed against CD81 (Novus Biologicals, Centennial, CO) and CD63 (Novus Biologicals) under non-reducing conditions and against CaM (Abcam, Cambridge, UK), Alix (Cell Signaling Technology), HSP70 (Enzo Life Sciences, Farmingdale, NY, USA), and GAPDH (Fujifilm Wako Pure Chemical Corporation, Osaka, Japan). Membranes were washed with TBS-T (TBS supplemented with 0.1% Tween 20) 4 times and incubated in Can Get Signal Immunoreaction Enhancer Solution 2 (Toyobo) containing HRP-conjugated secondary antibodies (anti-mouse IgG and anti-rabbit IgG, Cell Signaling Technologies) at a 1:16,000 dilution at room temperature for 1 h. Proteins were visualized by chemiluminescence using Clarity Western ECL Substrate (Bio-Rad, Hercules, CA, USA) and Light Capture II (Atto).

### 2.6. Immunocytochemical Staining

A total of 3 × 10^4^ cells of T-REx AspALP were plated on gelatin-coated coverslips and cultured in DMEM supplemented with 10% fetal bovine serum. Extracellular vesicles were removed by centrifugation at 11,000× *g* for 24 h followed by incubation in 1 μg/mL doxycycline and Penicillin-Streptomycin with or without 20 μM W13 at 37 °C for 72 h in 5% CO_2_/95% humidified air. Cells were fixed with 4% paraformaldehyde (FUJIFILM Wako Pure Chemical Corporation) at room temperature for 10 min and washed with PBS 3 times. Cells were blocked in PBS supplemented with 1% bovine serum albumin (BSA), 10% normal goat serum (NGS), 0.1% Triton-X, and 0.05% sodium azide at room temperature for 30 min. Cells were incubated with primary antibodies diluted in PBS supplemented with 1% BSA, 10% NGS, and 0.05% sodium azide at a 1:200 dilution at room temperature for 1 h. Primary antibodies were directed against CD63 (Novus Biologicals), LAMP2 (Bioss Antibodies Inc., Woburn, MA, USA) and CaM (Abcam). Cells were then washed with PBS 3 times and incubated in PBS supplemented with 1% BSA, 10% NGS, and 0.05% sodium azide containing Alexa488-conjugated anti-mouse IgG (Thermo Fisher Scientific) at a 1:500 dilution, Alexa546-conjugated anti-rabbit IgG (Thermo Fisher Scientific) at a 1:500 dilution, and 5 μg/mL Hoechst33342 (Sigma-Aldrich) at room temperature for 30 min. Cells were washed with PBS 3 times and mounted on glass slides with ProLong Diamond mounting medium (Thermo Fisher Scientific). Images of cells were taken using a BZ-X710 fluorescent microscope equipped with a confocal system (Keyence, Osaka, Japan).

### 2.7. Immunoprecipitation

Cell pellets from T-REx Mock and T-REx AspALP cells were lysed in Cell Lysis Buffer M (FUJIFILM Wako Pure Chemical Corporation) supplemented with Protease Inhibitor Cocktail Set V (EDTA free) (FUJIFILM Wako Pure Chemical Corporation) at 1 × 10^7^ cells/mL. Dynabeads protein A (Thermo Fisher Scientific) was mixed with anti-CaM antibodies (Abcam) and formed the magnetic bead-antibody complex. The complex was mixed with 200 μL of cell lysates in the presence or absence of 10 mM EGTA with gentle rotation. The magnetic bead-antibody-antigen complex was washed 3 times. Regarding Western blotting, the complex was supplemented with 30 μL of SDS loading buffer (Cell Signaling Technologies) and denatured at 95 °C for 5 min. In the MS analysis, the complex was lysed in 100 μL of 8 M urea.

### 2.8. MALDI-TOF-MS

Exosomes (100 μL each) and immunoprecipitants (5 μL each) were dissolved in 100 μL of 8 M urea. Peptides were concentrated from the solution with GL-Tip SDB and GC columns (GL Sciences Inc., Tokyo, Japan) and eluted with 20 μL of 80% acetonitrile supplemented with 0.1% trifluoroacetate. The peptide solution was mixed at 1:1 with 10 mg/mL CHCA in 50% acetonitrile supplemented with 0.1% trifluoroacetate, and 1 μL of the mixture was plated on an MTP384 target plate ground steel (Bruker, Billerica, MA, USA). After the plate was air-dried, peptides were measured using an ultrafleXtreme MALDI-TOF MS (Bruker) and analyzed by Flex analysis software (Bruker). In the MS/MS analysis, amino acid sequences were elucidated within an error of 0.7 Da.

### 2.9. Statistical Analysis

Statistical analyses were performed using a two-tailed *t*-test or one-way ANOVA with Tukey’s post hoc tests. Differences were considered to be significant when the *p*-value was less than 0.05. Each number of experiments was performed at least three times.

## 3. Results

### 3.1. Effects of W13, a CaM Inhibitor, on T-REx Mock and T-REx AspALP Cells

One function of CaM is to regulate the cell cycle and W13, a CaM inhibitor, reversibly delays the transition to the S phase in CHO-K1 cells [19]. To confirm that W13 inhibited CaM functions in T-REx Mock and T-REx AspALP cells, we examined the number and viability of cells in the presence and absence of W13 (Figure 1). When cultured in the presence of W13 for 72 h, T-REx Mock and T-REx AspALP cells remained sparse and mostly unchanged from the start of the culture (Figure 1A). Furthermore, the number of each cell type was significantly lower in the presence of W13 (Figure 1B). However, no significant difference was observed in cell viability (Figure 1C) and Annexin V labeling (Figure 1D), indicating that W13 functionally inhibited the proliferation of both cells. To examine whether W13 affected the translation of SEAP, activity was measured in the presence or absence of W13 (Figure 1E). Although the number of cells decreased, no significant difference was noted in SEAP activity.

### 3.2. Effects of W13 on the Signal Peptide Content in Exosomes from T-REx AspALP Cells

To examine alterations in the C-terminal fragment of APP SP in exosomes, the peak size at *m/z* 861 in the exosomal peptide fraction was analyzed by mass spectrometry (Figure 2A,B). The peak size at *m/z* 861 was detected in exosomes from T-REx AspALP cells in the absence of W13 but was reduced in exosomes from W13-treated cells to the same level as that in exosomes from T-REx Mock cells. The MS/MS analysis identified the peak at *m*/*z* 859 ± 4 as the C-terminal fragment of APP SP (LAAWTARA), which contained a -TAR-amino acid sequence (Figure 2C). To confirm the properties of exosomes from T-REx Mock and T-REx AspALP cells in the presence of W13, the average size and number of released exosomes were assessed by the NTA. No significant differences were observed in the average size of released exosomes in the presence and absence of W13 (Figure 2D). The number of released exosomes per T-REx cell increased in the presence of W13 (Figure 2E). These results indicated that CaM participated in the transport of SP fragments to exosomes.

### 3.3. Changes in Exosomal Cargo Protein Abundance in T-REx Cells in the Presence or Absence of W13

Since exosomes are secreted via endosomes, the distribution of CaM and CD63, which are markers of multivesicular endosomes and exosomes, in T-REx AspALP cells was examined in the presence or absence of W13 (Figure 3A). The co-localization of CaM and CD63 was noted in T-REx AspALP cells in the absence of W13 but appeared to increase in the presence of W13 in addition to increases in their expression. In addition, the distribution of CD63 and LAMP2, which is a marker of the lysosome, was examined in the presence or absence of W13 (Figure 3B). The colocalization of CD63 and LAMP2 was increased in the presence of W13, indicating degradation may be induced partly. To clarify changes in exosomal cargo protein abundance in T-REx cells in the presence or absence of W13, the expression of cargo proteins, such as CaM, Alix, HSP70, CD63, and CD81, was examined by Western blotting (Figure 3C). The expression of HSP70 remained unchanged in the presence of W13, whereas that of cargo proteins was significantly up-regulated. These results showed that the inhibition of CaM induced increases in some cargo proteins in T-REx cells.

### 3.4. Cargo Proteins in Exosomes from T-REx Cells

To clarify changes in cargo proteins in exosomes from T-REx cells, cargo proteins in exosomes collected from the extracellular fluid of these cells were examined (Figure 4). The content of CaM remained unchanged in the presence of W13, whereas that of other cargo proteins, such as Alix, HSP70, CD63, and CD81, decreased. These results indicated the involvement of CaM in the transport of some cargo proteins to exosomes.

### 3.5. Interactions between CaM, Signal Peptides, and HSP70 and CD81 in T-REx Cells

To establish whether CaM bound to APP SP in T-REx cells, a peptide fraction extracted from immunoprecipitated samples with the CaM antibody was analyzed by MALDI-TOF-MS (Figure 5A,B). The peptide fraction from T-REx AspALP cells in the absence of W13 contained APP SP, whereas that in the presence of W13 did not. Furthermore, peptide fractions extracted from EGTA-added immunoprecipitants did not contain APP SP. To clarify whether other factors were contributing to the CaM-dependent transport of SPs into exosomes in T-REx cells, immunoprecipitants were analyzed by Western blotting (Figure 5C). Low levels of HSP70 and CD81, but not CD63 or Alix, were detected in the immunoprecipitants. In addition, elevated levels of HSP70 and CD81 were noted in the immunoprecipitants of W13-treated T-REx cells, and these increases were suppressed by the treatment of immunoprecipitants with EGTA, a calcium chelator. 

## 4. Discussion

We herein demonstrated for the first time that the encapsulation of SP fragments into exosomes was inhibited in the presence of a CaM inhibitor, which was consistent with our previous findings showing that SP fragments bound to CaM in a cell-free system [12,13]. The binding of CaM to the APP SP fragment was increased by the addition of Ca^2+^ in our previous studies, while the APP SP fragment in CaM immunoprecipitants from T-REx AspALP cells was decreased by the treatment of immunoprecipitants with EGTA, a Ca^2+^ chelator, in the present study (Figure 5B). Since CaM is distributed in exosomes and in the ER where SPs are produced, it is considered to play an important role in the transport of SPs from the ER to exosomes. Although CaM is an exosomal component [15], its functions remain unclear. It has been shown to bind to a wide variety of peptides [20]; however, limited information is currently available on the binding of CaM to SPs or their fragments [12,13]. CaM participates in the transduction of multiple signals for many crucial processes and translocates among the plasma membrane, cytoplasm, and organelles containing endosomes [14,21]. Therefore, the present results suggest that CaM binds to SP fragments produced in the ER and translocates to endosomes, and also that CaM-SP fragment complexes are encapsulated into exosomes. Cells generally have many proteases and peptidases, and unnecessary proteins and peptides are intracellularly degraded by their enzymes. The binding of CaM to SPs or SP fragments may play a role in protection against peptide degradation as well as intracellular peptide transport. Some SPs and SP fragments have physiological functions and are released into extracellular fluid via exosomes. Exosomes are extracellular vesicles for intercellular communication and only cells that receive exosomes display functions derived from exosomal components. Therefore, exosomal SPs and SP fragments may be involved in the functional regulation of specific cells that receive exosomes. 

The CaM inhibitor W13 has been shown to induce the sustained activation of ERK2 and expression of p21 (cip1) in NIH 3T3 fibroblasts, which results in the inhibition of cell proliferation [22]. In the present study, the proliferation of T-REx Mock and T-REx AspALP cells was also inhibited in the presence of W13 (Figure 1A–C) and the number of exosomes released per cell simultaneously increased (Figure 2B). A recent study reported similar findings; inhibitors of cell proliferation, cucurbitacin B, gossypol, and obatoclax, promoted the secretion of extracellular vesicles [23,24,25,26]. Therefore, the inhibition of proliferation by W13 may have enhanced exosome release. Previous studies identified CaM as a component of exosomes [15,27]; however, its role was unknown. W7, a CaM inhibitor, has been shown to increase the exocytosis of CD63-positive vesicles in human neutrophils [28]. These findings suggest the involvement of CaM in the release of granules/vesicles, including exosomes. CaM dysfunction by the W13 treatment induced the abnormal colocalization and accumulation of CD63 and CaM in T-REx AspALP cells (Figure 3A). Since CD63 is specifically distributed in endosomes and exosomes [29], CaM may accumulate in endosomes. Moreover, the abundance of CD63 increased in W13-treated T-REx cells (Figure 3B). A dysfunctional endosomal pathway and abnormally numerous and enlarged early endosomes in neurons have been detected in Alzheimer’s disease and Down syndrome [30,31]. In addition, enhanced exosome secretion and increased CD63 levels have been reported in the brains of Down syndrome patients [32]. These findings indicate that the up-regulated expression of CD63 promotes exosome release as an endogenous mechanism mitigating endosomal abnormalities in Down syndrome. Since CaM dysfunction by the W13 treatment displayed similar results, CaM may play an important role in physiological endosomal functions. When T-REx cells were treated with W13, the intracellular abundance of many cargo proteins, such as CaM, Alix, CD63, and CD81, increased (Figure 3B). On the other hand, the number of cargo proteins, such as Alix, HSP70, CD63, and CD81, in exosomes decreased (Figure 4). These results indicate that CaM dysfunction in W13-treated T-REx cells inhibited the transport of many cargo proteins to exosomes; therefore, CaM may be involved in cargo protein carrying and loading into exosomes. In fact, several groups have shown that W13 treatment affects membrane trafficking of epidermal growth factor receptors and transferrin [33,34]. CaM is associated with endosomal fusion [35], and W13 treatment inhibits the exit of several proteins from early endosomes. Therefore, endosomal dysfunction caused by W13 treatment may have inhibited cargo proteins and SP transport.

CaM immunoprecipitants from T-REx AspALP cell lysates contained the APP SP fragment and disappeared following the W13 treatment (Figure 5A,B). HSP70 and CD81 were also found in immunoprecipitants (Figure 5C). However, HSP70 and CD81 increased in CaM immunoprecipitants from W13-treated cells (Figure 5C). These results suggest that the binding sites for CaM differ between SPs and HSP70 or CD81. In other words, W13 may compete with the binding site of SPs, resulting in an increase in the amount of binding to HSP70 and CD81. Moreover, we found that the CaM complex between HSP70 and CD81 in W13-treated cells dissociated in the presence of EGTA, a calcium chelator (Figure 5C). The interaction of CaM with HSP70 was also shown to dissociate in the presence of EGTA [36]. Since CaM binds to SP fragments in a Ca^2+^-dependent manner [12], the interactions among CaM, HSP families, CD81, and SP fragments may be important for the transport of SP fragments into exosomes. HSP70 is a central component of the cellular network of molecular chaperones and folding catalysts. It assists in a large number of protein-folding processes in cells by the transient association of their substrate-binding domain with short hydrophobic peptide segments within their substrate proteins [37]. Although the binding of HSP70 to CaM has been demonstrated in plants [36,38,39], the relationship between CaM and HSP70 also regulates the cell cycle in human cells [16]. HSP70 interacts with other HSP family members, such as HSP40 and HSP90 [40], and HSP90 binds to CaM and displays several functions [41,42]. Although CaM and HSP family members are both exosomal components, it currently remains unclear whether their interaction is involved in transport to exosomes. Another E-F hand protein, ALG-2, is also a constituent protein of exosomes and is known to mediate the interaction between ALIX and TSG101, which constitutes the endosomal sorting complexes required for transport machinery-I (ESCRT-I), in a Ca^2+^-dependent manner [43]. Therefore, the interaction between CaM and HSP families may contribute to the transport of cargo proteins into exosomes. On the other hand, CD81 is a tetraspanin and marker molecule for exosomes and CD63 [44]. Tetraspanins are involved in biological processes, such as cell adhesion, motility, invasion, membrane fusion, signaling, and protein trafficking [45]. Different tetraspanin members have been shown to regulate protein sorting into exosomes. Breast cancer-associated fibroblast-derived exosomes are enriched in tetraspanins, such as CD63, CD81, and CD82, whereas only CD81 is responsible for the transport of Wnt 11 cargo to exosomes [46]. The amyloidogenic pigment cell-specific type I integral membrane protein is sorted into multiple vesicular endosomes by CD63 [47]. Although there is currently no direct evidence to show the interaction/binding of CaM with CD63 or CD81, CaM has been shown to bind to one of the tetraspanins, peripherin/rds, which is present in both rod and cone photoreceptor cells [48]. Therefore, the relationship between CD81 and CaM may be important for the sorting of SPs into exosomes in T-REx AspALP cells.

Exosomal biogenesis involves two pathways: ESCRT-dependent and -independent pathways [49]. Although the mechanisms by which the CaM-SP fragment complex is involved in the ESCRT-dependent or -independent pathway have not yet been elucidated, CaM played an essential role in wound repair of the cell membrane through interactions between CaM and ESCRT components in *Dictyostelium* cells [50]. These findings indicate that the CaM-SP fragment complex interacts with ESCRT components in exosomal biogenesis, which warrants further study.

In conclusion, CaM functions as a key regulator of the transport of SP fragments into exosomes. It associates with cargo proteins, such as HSP70 and CD81, which contributes to the transport of cargo proteins as well as SP fragments into exosomes. The present results demonstrate that CaM has novel roles in the sorting of contents during exosomal biogenesis.

## Figures and Tables

**Figure 1 cells-12-00158-f001:**
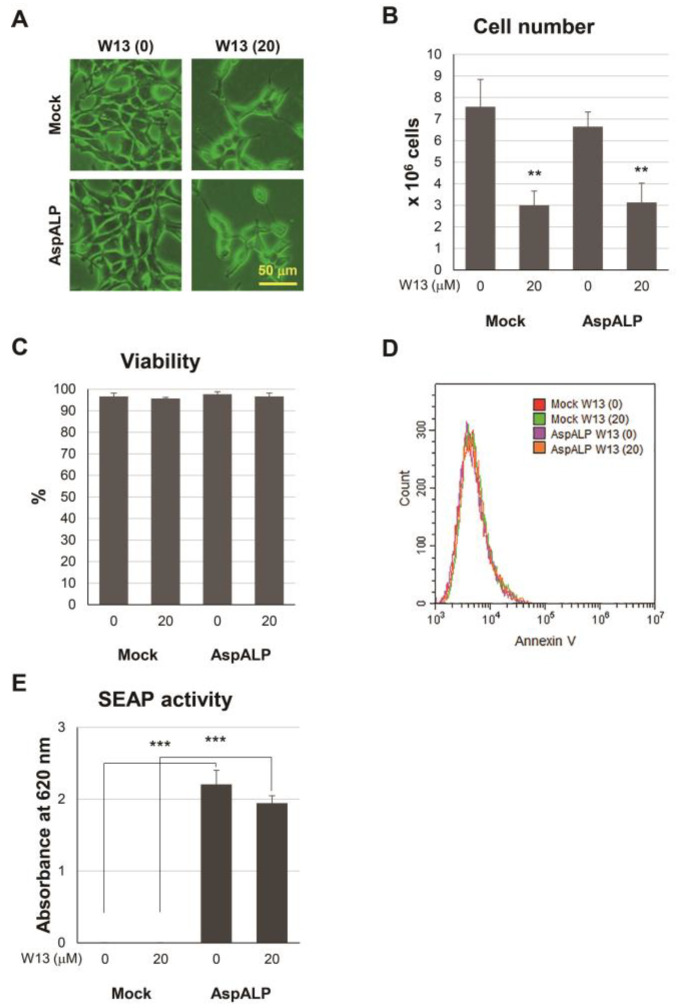
Properties of T-REx Mock and T-REx AspALP cells in the absence or presence of W13, a CaM inhibitor. (**A**) Representative images of T-REx Mock (Mock) and T-REx AspALP (AspALP) cells at 0 or 20 μM W13 for 72 h. (**B**) The number of Mock and AspALP cells at 0 or 20 μM W13 for 72 h. ** *p* < 0.01 vs. Mock cells at 0 μM W13. (**C**) The viability of Mock and AspALP cells at 0 or 20 μM W13 for 72 h. (**D**) Annexin V-labelled Mock and AspALP cells at 0 or 20 μM W13 for 72 h. (**E**) The SEAP activity of Mock and AspALP cells at 0 or 20 μM W13 for 72 h. *** *p* < 0.001 vs. Mock cells.

**Figure 2 cells-12-00158-f002:**
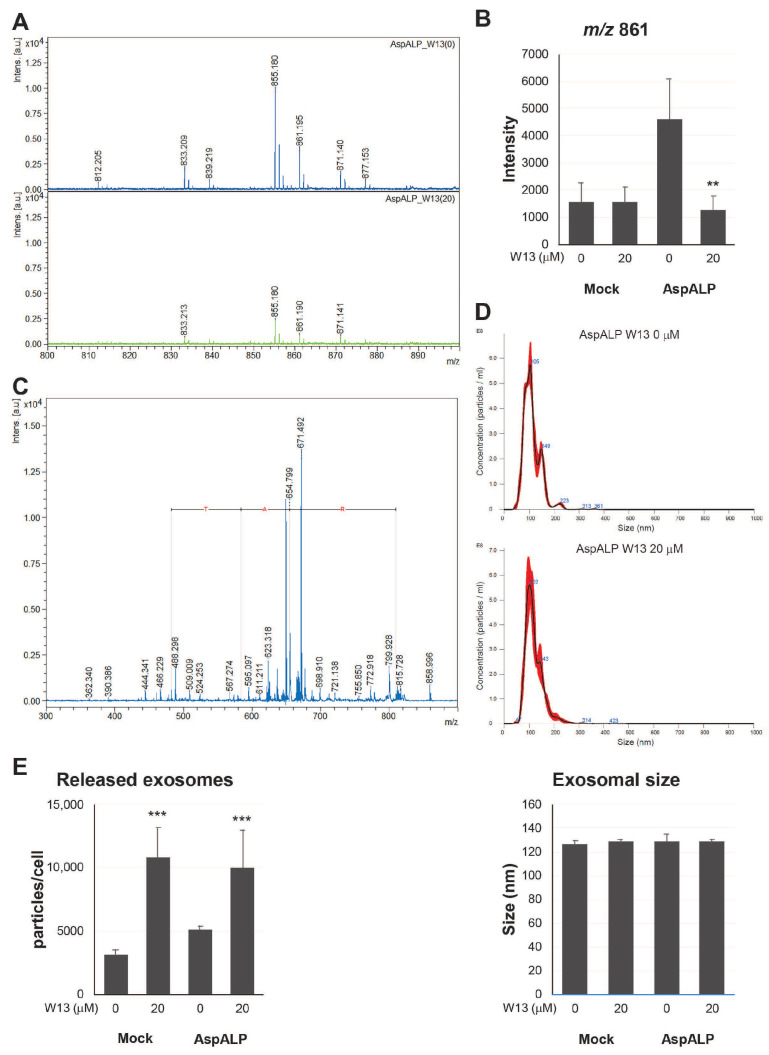
Properties of exosomes from T-REx cells in the absence or presence of W13. (**A**) The peptide solution extracted from the exosomes of T-REx AspALP (AspALP) cells at 0 or 20 μM W13 for 72 h was analyzed by MALDI-TOF-MS. (**B**) The peak intensity at *m/z* 861 was measured by MALDI-TOF-MS. ** *p* < 0.01 vs. AspALP cells at 0 μM W13. (**C**) MS/MS analysis of peaks at *m/z* 861 ± 4. (**D**) The upper and middle panels show representative profiles of exosomes from AspALP cells at 0 or 20 μM W13 for 72 h. The lower panel shows the average size of exosomes released from T-REx Mock (Mock) and AspALP cells at 0 or 20 μM W13 for 72 h. (**E**) Released exosomes per cell. *** *p* < 0.001 vs. Mock cells at 0 μM W13.

**Figure 3 cells-12-00158-f003:**
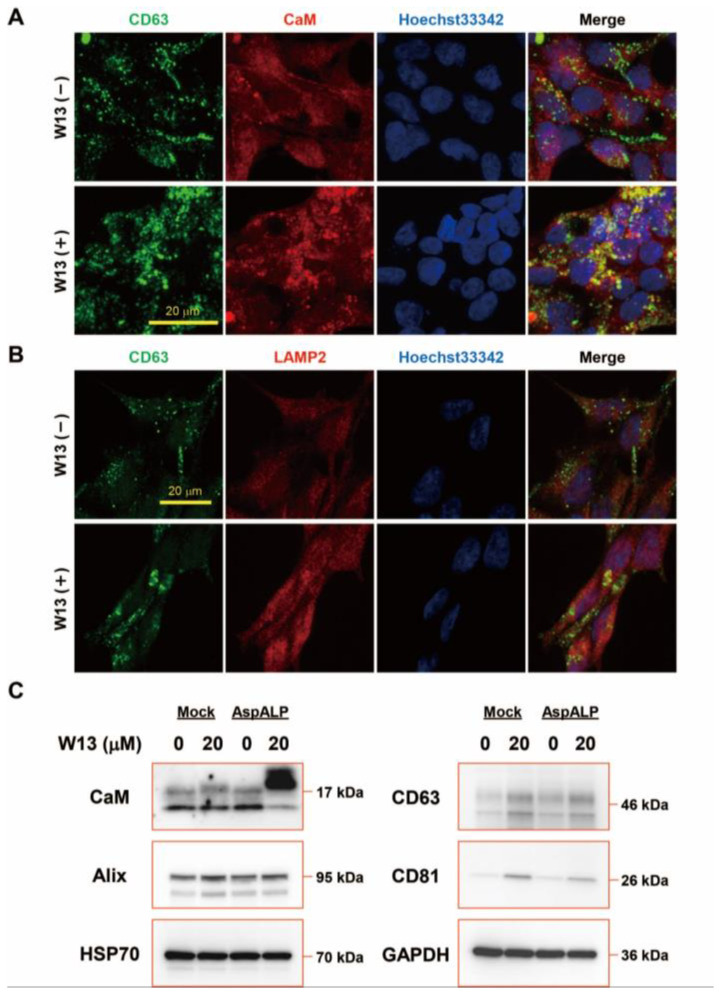
Exosomal cargo proteins in T-REx cells in the absence or presence of W13. (**A**) Distribution of CD63 and CaM in T-REx AspALP cells in the absence (−) or presence (+) of W13 for 72 h. (**B**) Distribution of CD63 and LAMP2 in T-REx AspALP cells in the absence (−) or presence (+) of W13 for 72 h. (**C**) Western blot analysis of exosomal cargo proteins from T-REx cells in the absence or presence of W13 for 72 h. Representative images from three independent experiments.

**Figure 4 cells-12-00158-f004:**
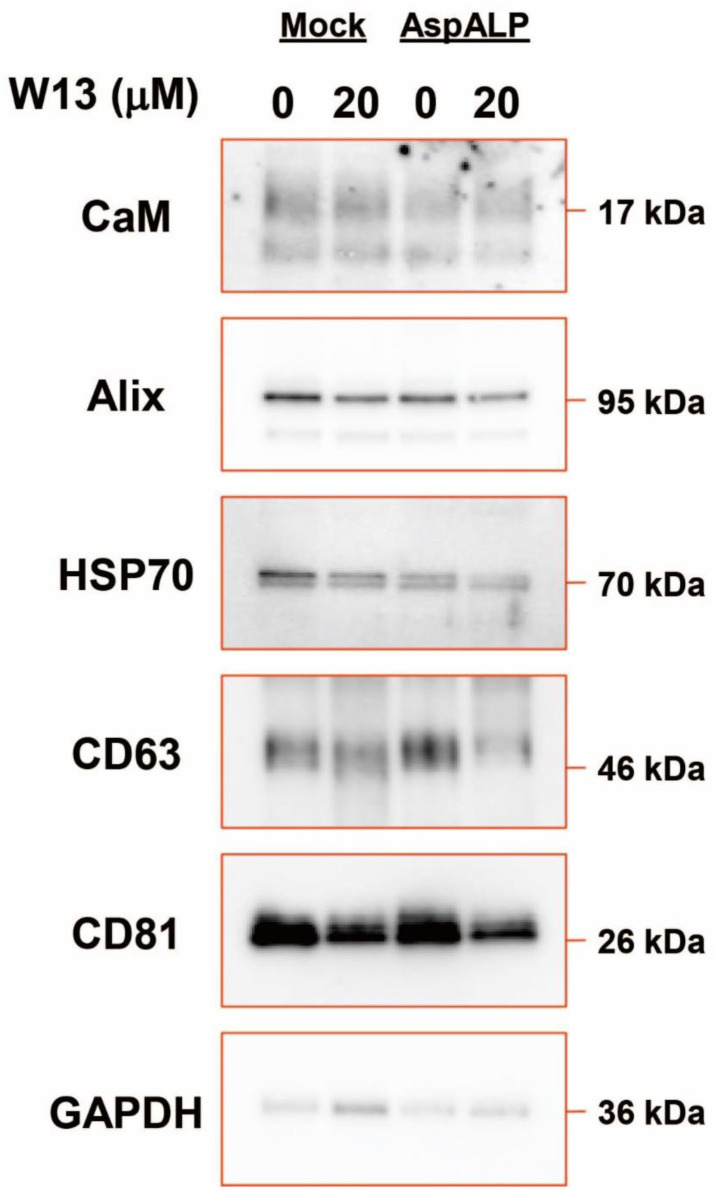
Cargo proteins in exosomes from T-REx cells in the absence or presence of W13. Western blot analysis of cargo proteins from T-REx cells in the absence or presence of W13 for 72 h. An equal amount of protein was loaded. Representative images from three independent experiments.

**Figure 5 cells-12-00158-f005:**
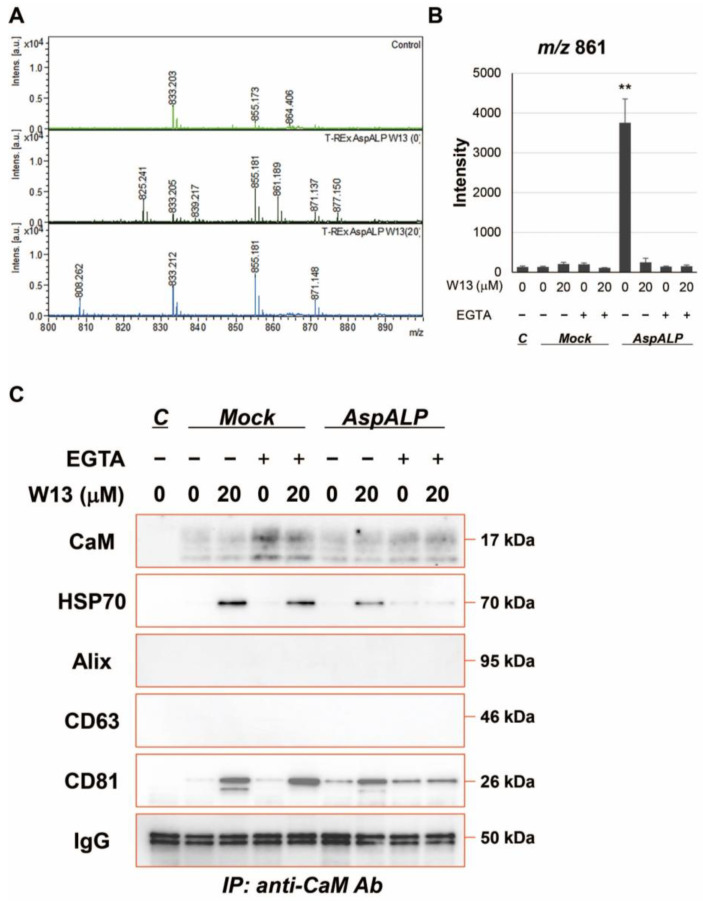
Interaction of CaM with HSP70 and CD81 in T-REx cells (**A**) The peptide solution extracted from immunoprecipitants with the CaM antibody from T-REx cells was analyzed by MALDI-TOF-MS. (**B**) The peak intensity at *m/z* 861 was measured by MALDI-TOF-MS. ** *p* < 0.01 vs. Mock cells in the absence of W13 and EGTA. C; control, Mock; T-REx Mock, AspALP; T-REx AspALP. (**C**) Western blot analysis of immunoprecipitants with the CaM antibody from T-REx cells. Representative images from three independent experiments.

## Data Availability

The raw data can be provided by the corresponding author upon reasonable request.

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
