# Peer review of "Calmodulin as a Key Regulator of Exosomal Signal Peptides"

_cells, 2022, doi:10.3390/cells12010158_

Round 1

Reviewer 1 Report

Comments to the authors

Summary:

The present manuscript, which is a continuity of recently published articles by the same authors, reveals very interesting results for a new role of calmodulin in exosome loading and biogenesis. In particular, the study is focused in the involvement of calmodulin in signal peptide fragments (SPs) loading of into exosomes. This study, by using T-REX AspALP cells expressing APP (amyloid precursor protein) SP (APPSP), describes that inhibition of calmodulin by its specific inhibitor W13 impairs or diminishes the presence of APPSP fragments in exosomes and this effect is also extensive for other exosome cargoes like Alix, CD63, CD81 or HSP70.

I am in principle supportive of accepting this work for publication. However, from my point of view several questions need to be clarified in order for the manuscript to be suitable for publication.

Comments and suggestions for the Authors:

1) From the point of view of this reviewer, the authors’ conclusions given in the text for the work presented in this study are too strong and unwarranted from the results shown in the article. They are not completely supportive of a direct effect of calmodulin, by interacting with SP fragment, in the loading of SP fragments into the secreted exosomes. Therefore, I think that the conclusions of the article have to be softened. It is well known the pleiotropic effects of calmodulin in several cellular processes.

Several published articles have demonstrated that inhibition of calmodulin, using its specific inhibitor W13 or W7, affects trafficking of EGFR and other proteins, and molecules like dextran, in the endocytic compartment. W13 inhibits the exit from early endosomes and consequently produces aberrant enlarged endosomes. It is plausible that W13, by interfering the regular traffic of proteins in such complex endocytic compartment, could affect exosome biogenesis in the LE/MVBs (Late endosome/multivesicular bodies). Thus, this W13 effect could accumulate exosome cargoes intracellularly and decrease them into released exosomes, which it is reported in the present manuscript.

My concern is that the effect that the authors describe in this study could be a consequence of a general inhibition of endocytic trafficking elicited by W13 treatment instead of a specific inhibition of SPs-calmodulin complexes that could avoid SPs exosomal loading.

 I agree that this issue is not easy to elucidat but at least this possibility have to be mention and discussed in the text and also added the references behind W13 effects on endocytic trafficking.

2) Based on published article (reference 14), authors assume that inhibition of calmodulin with W13 delays the transition to the S phase. In figure 1, it is shown that W13 treatment decreases proliferation of each cell line tested but without changes in their cell viability. Please, could the authors better explain how cellular viability was measured in this experiment?

Please, could the authors analyze cell cycle in control and W13-treated cells by flow cytometry (e. BrdU and/or annexinV labelling) in the experimental conditions performed in their experiments? I think this is necessary to understand and discard any speculation about other possible origins of extracellular vesicles (for instance due to apoptosis) than exosomes measured by NTA (nanoparticle tracking analysis) in the present study.

3) In figure 2, it is shown that after W13 treatment there was no difference of exosomes size measured by NTA but the amount of exosomes secreted increased importantly.

Please, could the authors also show the NTA EVs profile in figure 2? This is important to better evaluate if there are any presence of other particles not related with exosomes (e. protein complexes, possible vesicles from dead apoptotic cells…) that increased the number of particles, maybe erroneously count as exosomes, after W13 treatment.

Sincerely, I cannot understand why W13 increases the number of secreted exosomes but these exosomes have less amount of known markers/cargoes of them (Alix, HSP70, CD63 and CD81) (figure 4). It is necessary to probe that these purified exosomes do not contain contamination of different intracellular compartments. Please, could you re-probed these membranes with   markers of Golgi, ER (e. calreticulin, calnexin) or tubulin, which theoretically should be absent in exosomes?

Minor points:

Please, explain the different bands of calmodulin observed in the western blot in panel B of figure 3.

Please, could you add the molecular weight markers in western blots?

Reviewer 2 Report

Review report:

The aim of the paper is to explore the role of calmodulin as key regulator in transporting of signal peptide (SP) fragments into exosomes. The finding that these signal peptides could influence sorting of contents into exosomes is very interesting as it furthers the current mechanistic understanding of the exosome biogenesis and cargo sorting.

Highlighting calmodulin as a potential regulator of SP sorting into exosomes is interesting, however this has also been shown in the group’s previous publications taking away the novelty aspect from the paper. The work here verifies their previously finding within the cell system rather than in cell-free system used previously. While the downstream role of these signal peptides in exosome biogenesis has been speculated in the discussion its role has not necessarily been verified. But I understand this is not in the remit of the paper.

General concept comments:

Experiments have been performed in presence of W13 an inhibitor of calmodulin to highlight the role of calmodulin in signal peptide sorting. Using another independent CaM inhibitor to ensure no non-specific effect of W13 would further verify the CaM function further.

Encapsulation of SP fragments into exosomes was increased in presence of W13, would opposite be the case with CaM agonist or addition of calcium. As previous work was performed in cell-free system, would same be the case within cell system?

The work here is performed in a system which overexpresses APP SP, can the authors address (at least in part in discussion), the limitation of using such a system. Are you pushing the system to incorporate these signal peptides into exosomes simply by overexpressing?  

Since it is suggested that W13 treatment decreases incorporation of cargo proteins such as Alix, Cd63 (Fig 4) but their overall expression within cells are increased (Fig 3), are these cargoes pushed to other cellular pathway as a result such as degradation. Studying colocalization with lysosomal markers for instance in Fig 3 would provide further clarity.

It is still unclear if the APP SP interacts directly with CaM or to other cargo proteins such as CD81 and/or Hsp70.

Specific comments:

The n numbers for the experiments are not clear.

Length of W13 treatment in all the experiments are not clear, 72h for all the experiments.

For imaging and western blot data (Fig 3,4 and 5), number of repeats are not clarified, i.e. representative images from x number of independent experiments.

Quantification of the data would be helpful for Fig 3A showing colocalization.

For western blot images (Fig 3B and 4), please clarify if both bands represent the protein of interest as it is not clear and have the molecular weights alongside.

Result section 3.4 (referring to fig 4), CaM blots are not very clear. As CaM being an exosomal component underlies the study, a clearer blot showing CaM bands would be needed. Here please also clarify how the exosomes were normalised i.e. if equal number of exosomes were loaded or equal amount of protein was loaded?
